# Prenatal Diagnosis of Chromosome 16p11.2 Microdeletion

**DOI:** 10.3390/genes13122315

**Published:** 2022-12-08

**Authors:** You Wang, Hang Zhou, Fang Fu, Ken Cheng, Qiuxia Yu, Ruibin Huang, Tingying Lei, Xin Yang, Dongzhi Li, Can Liao

**Affiliations:** 1The First School of Clinical Medicine, Southern Medical University, Guangzhou 510515, China; 2Department of Prenatal Diagnostic Center, Guangzhou Women and Children’s Medical Center, Guangzhou Medical University, Guangzhou 510620, China; 3School of Medicine, South China University of Technology, Guangzhou 510641, China

**Keywords:** 16p11.2 microdeletion, CMA, prenatal diagnosis, genetic counseling

## Abstract

(1) Objective: To investigate the prenatal diagnosis and genetic counseling for 16p11.2 microdeletion syndrome and to evaluate its pregnancy outcome. (2) Methods: This study included 4968 pregnant women who selected invasive prenatal diagnoses from 1 January 2017 to 1 August 2022. These 4698 pregnancies underwent chromosomal microarray analysis (CMA), data on 81 fetuses diagnosed with 16p11.2 microdeletion syndrome based on prenatal ultrasound features and genetic test results were recorded, and their pregnancy outcome was evaluated. (3) Results: 1.63% of fetuses (81/4968) were diagnosed with 16p11.2 microdeletion syndrome. Among these, there were skeletal malformations in 48.15% of the 81 fetuses, cardiovascular malformations in 30.86%, central nervous system malformations (CNS) in 11.11%, digestive system structural abnormalities in 6.17%, and isolated ultrasonography markers in 3.70%. (4) Conclusions: 16p11.2 microdeletion syndrome can display various systemic ultrasound abnormalities in the perinatal period but vertebral malformations are the most common. Our study is the first to report that *TBX1* and *CJA5* are associated with 16p11.2 microdeletion syndrome, expanding the disease spectrum of 16p11.2 microdeletion syndrome. In our study, the ventricular septal defect is the main feature of cardiac structural abnormalities caused by 16p11.2 microdeletion syndrome. In addition, our study highlights the use of CMA in 16p11.2 microdeletion syndrome, analyzed their genetic results, and evaluated the follow-up prognosis, which can be useful for prenatal diagnosis and genetic counseling.

## 1. Introduction

The 16p11.2 microdeletion (OMIM: 611913) phenotype is a condition caused by a recurrent heterozygous deletion region on chromosome 16 with a prevalence of 0.03% [1]. The condition is often referred to as 16p11.2 microdeletion syndrome. The definition of a 16p11.2 microdeletion phenotype was first proposed by Kumar et al. in 2008 [2]. The main clinical manifestations are intellectual disability, developmental delay, epilepsy, being overweight, and vertebral deformities, but vertebral deformities are the most common [3]. The clinical features include psychiatric/behavioral issues present in more than 90% of cases. This CNV has an incomplete penetrance and variable expressivity showing interindividual and intrafamily variability. The condition is often associated with cognitive, neurological, or psychiatric functional abnormalities [4,5], thus it is a huge burden from families to society. Typical deletions range from 500 to 600 kb and involve 24 to 29 genes, with a prevalence of approximately 1/2000 in the population and up to 0.5% in children with autism [6]. 16p11.2 microdeletion refers to the ~600 kb heterozygous deletion between BP4 and BP5 between 29.5 and 30.1 Mb on chromosome 16 in the reference genome (GRCh37/hg19) [7,8,9]. This region is flanked by 147 kb low copy repeat (LCR) sequences with 99.5% sequence identity [8]. The genomic instability caused by two low-copy repetitive sequences makes it susceptible to non-allelic homologous recombination (NAHR) on homologous chromosomes during meiosis, leading to microdeletions [10]. 16p11.2 microdeletion (OMIM: 611913, NCBI36) distal breakpoints range from 29,427 to 29,591 kb, while proximal breakpoints range from 30,177 to 30,344 kb. 16p11.2 microdeletion syndrome can be detected by chromosomal microarray analysis (CMA) or multiplex ligation probe amplification (MLPA) [9,11,12]. A large portion of 16p11.2 microdeletions, up to 93%, appear to arise de novo. This means they are rarely inherited from parents, and it is particularly rare that they occur in multiple generations in the same family [8]. At present, little has been reported about a prenatal diagnosis of 16p11.2 microdeletion syndrome. Our study aimed to evaluate the use of CMA in 16p11.2 microdeletion syndrome, which, combined with ultrasound abnormalities, contributes to a better understanding of the intrauterine phenotype associated with this microdeletion and provides a basis for prenatal diagnosis and genetic counseling of the fetus.

## 2. Materials and Methods

The enrollment criteria in our study: (1) 4968 pregnant women who underwent invasive testing between 1 January 2017 and 1 August 2022; (2) All 4968 samples obtained by invasive diagnosis were tested by CMA. In our study, the clinical indications for CMA: (1) a history of recurrent miscarriage; (2) a history of adverse births such as growth retardation and multiple malformations; (3) prenatal ultrasound detection of fetal structural abnormalities. The average age of pregnant women was 30.1 years (19–47 years) and the average gestational weeks was 25.8 weeks (12–37 weeks). The invasive procedure for prenatal diagnosis is determined by the week of gestation and a sample of 20 mL of fetal amniotic fluid is collected by amniocentesis at less than 25 weeks of gestation. A fetal cord blood sample of 5 mL was collected by cord blood puncture at gestational weeks greater than 25 weeks or if oligohydramnios was present. The combination of two or more ultrasound markers is called a non-isolated group, and only one ultrasound marker is called an isolated group. Ultrasonic markers include soft markers and structural abnormalities. All pregnant women received detailed genetic counseling and signed an informed consent form before invasive prenatal diagnosis. The Medical Ethics Committee approved the study of Guangzhou Women’s and Children’s Medical Center.

Invasive samples were analyzed using the following methods. Quantitative fluorescent polymerase chain reaction (QF-PCR) to screen for aneuploidy on chromosomes 13, 18, 21, X, and Y or to exclude contamination of maternal cells by using MLPA kits (Guangzhou Darui Biotechnology Co., Ltd., Guangdong, China). CMA was canceled if the QF-PCR result indicated aneuploidy; in the case of trisomy 13 or 18, or 21, and monosomy X, cytogenetic analysis was carried out in its place. Karyotyping analysis was performed using conventional G-banding techniques (550-band resolution). Samples were subsequently subjected to CMA only when there was a normal QF-PCR result. CMA was performed using Affymetrix CytoScan HD/750K arrays with single nucleotide polymorphism arrays (SNP arrays) and array-based comparative genomic hybridization (aCGH) platforms according to the manufacturer’s protocol (Affymetrix Inc., Santa Clara, CA, USA), with resolutions of 10 and 100 kb. The constructed reference genome was aligned to GRCh37/hg19. According to the joint consensus recommendations of the American College of Medical Genetics (ACMG) and Clinical Genome Resources (ClinGen), copy number variant (CNV) are classified into five categories: pathogenic, likely pathogenic, variants of undetermined significance (VOUS), likely benign, and benign variants [9]. Only pathogenic and likely pathogenic variants were recorded in this study; likely benign and benign variants were not recorded. Where clinically significant variants or VOUS are found in samples from invasive procedures, in that case, parental CMA is recommended for these couples to clarify whether CNV is inherited or a de novo variant.

Statistical analysis was performed by using IBM statistical program SPSS 26.0. All qualitative data were expressed as *n* (%). The Chi-square test was used to assess the count data. *p* < 0.05 was statistically significant.

## 3. Results

### 3.1. Genetic Results

#### 3.1.1. Research Subjects

In our cohort, 4968 pregnant women underwent invasive prenatal diagnosis. Among them, 4879 pregnant women underwent QF-PCR; 4756 underwent chromosome karyotype analysis and 4968 underwent CMA testing at our prenatal diagnosis center from 1 January 2017 to 1 August 2022. The total detection rate of pathogenic/likely pathogenic variants is 14.1%, and the 22q11.2 deletion is the most frequent (5.6%), and then for deletion 16p11.2 (1.6%). The chromosomal microarray identified clinically significant variants including pathogenic copy number variants (pCNVs) in 75 fetuses and likely pathogenic CNVs in six fetuses. VOUS was found in 18 fetuses. Of these 81 pregnant women, 56 were multipara and 25 were primipara. The gestational age of invasive prenatal genetic testing was between 17 and 37 weeks. Amniocentesis was performed in 65 cases and percutaneous umbilical blood sampling was conducted in 16 cases. Table 1 summarizes the clinical details of 81 fetuses with 16p11.2 microdeletion in the study.

#### 3.1.2. Results of CMA

In those cases, detected with 16p11.2 deletion, eight chromosomal abnormalities were detected by QF-PCR. There were seven chromosomal numerical anomalies detected by QF-PCR including two cases with trisomy 21, two cases with trisomy 16, and one case with trisomy 13 confirmed by CMA. In 81 cases of 16p11.2 microdeletion, the CMA identified 1.50% (75/4968) of fetuses with pathogenic copy number variants (pCNVs) and 0.12% (6/4986) of fetuses with likely pathogenic CNVs. Compared with the population prevalence previously reported in the literature [8], the prevalence of 16p11.2 microdeletions in our cohort is relatively high at about 1.63%. Among 81 cases of 16p11.2 microdeletion, 71 cases proved to be a de novo variant. According to the CMA verification of parents, two cases were inherited from their mothers, one case was inherited from their father, and in seven cases we were unable to know the inheritance because their parents refused CMA testing. Considering economic factors, 11 couples in whom CNV was found VOUS in the fetus declined the offer of parental CMA testing for verification because its price was close to USD 1000. The 16p11.2 microdeletion refers to a recurrent ~600 kb heterozygous deletion between breakpoint 4 (BP4) and breakpoint 5 (BP5) on chromosome 16p11.2 between 29.5 and 30.1 Mb in the reference genome (GRCh37/hg19). The size of the 16p11.2 microdeletion in these 81 cases ranged from 587 kb to 918 kb. 16p11.2 microdeletion (OMIM: 611913, NCBI36) distal breakpoints range from 29,427 to 29,591 kb, while proximal breakpoints range from 30,177 to 30,344 kb [8]. These deletions were located in breakpoint 4 (BP4) (from 29,342 to 29,602 kb) and breakpoint 5 (BP5) (from 30,212 to 30,342 kb) [8].

### 3.2. Ultrasound Results

Table 2 summarizes abnormal ultrasound findings in 81 fetuses with 16p11.2 microdeletion. All of the 81 fetuses had ultrasound anomalies, 78 being isolated and three multiple. The 81 fetuses with deletion of CNVs in the 16p11.2 region had the following ultrasound findings: 48.15% (39/81) had skeletal malformations, 30.86% (25/81) had cardiovascular malformations, 11.11% (9/81) had CNS malformations, and 6.17% (5/81) had structural abnormalities of the digestive system. Moreover, 14.81% (12/81) had a combination of thickening nuchal translucency (NT) in addition to the above structural abnormalities, and 3.70% (3/81) had only one isolated ultrasound marker of the above abnormalities (Table 2). The detection rate of 16p11.2 microdeletion was statistically significant (*p* < 0.05) in fetuses with fetal skeletal malformations (48.15%, 39/81), fetuses with cardiovascular malformations (30.86%, 25/81) and fetuses with isolated ultrasound markers (3.70%, 3/81). In total, 48.15% of fetuses with skeletal malformations had vertebral anomalies, especially hemivertebra and butterfly vertebra. Other skeletal deformities included rib abnormalities, clubfoot, polydactyly, and ectrodactyly. Malformations of the cardiovascular system included ventricular septal defect (VSD), atrial septal defect (ASD), aortic stenosis, endocardial cushion defect, and pulmonary stenosis. Notably, four fetuses with structural malformations of the cardiovascular system additionally had skeletal malformations at 32–37 weeks, and all four cases exhibited hemivertebra malformations. In addition, isolated ultrasound markers such as widening of the lateral ventricles, permanent left inferior cavernous artery, and intense intraventricular spots were observed in 3.70% of fetuses. Some ultrasound markers were also present in a few cases of skeletal or cardiovascular malformations, including bilateral hydronephrosis, nasal dysplasia, femur/humerus shortening, and clubfoot.

### 3.3. Pregnancy Outcomes

Based on the genetic counseling and the couple’s decision, among all cases, 58 were selected for termination of the pregnancy but none agreed to have a post-induction autopsy. Table 3 summarizes the perinatal outcomes of our cohort. Three cases were delivered prematurely at 36 weeks, 35 + 2 weeks, and 36 + 3 weeks, respectively; 11 cases were normally delivered, and nine were by cesarean section. In the screening of 23 newborns, four cases of unilateral clubfoot and one case of bilateral clubfoot that were not detected by prenatal ultrasound were identified. Four cases of VSD and tetralogy of Fallot detected by prenatal ultrasound were confirmed after birth, and two cases received surgical treatment after birth, with a good prognosis. Eight newborns were confirmed to have hemivertebra deformities after birth and five developed scolioses in infancy; two were treated surgically at one year and two months and three years and five months of age, respectively, with a good prognosis. Two fetuses had combined skeletal and cardiac malformations. One fetus showed growth and psychomotor retardation and had a poor prognosis, while the other was treated surgically and is doing well.

## 4. Discussion

Weiss, L. et al. proposed that the prevalence of 16p11.2 microdeletion was 0.030–0.035% [4]. It was first identified as one of the most common genetic risk factors for autism spectrum disorders (ASD), with a prevalence of approximately 0.3–1.0% [4,5]. In addition, this deletion was associated with other psychiatric or neurological disorders (such as speech delay, mental retardation, developmental coordination problems, and epilepsy) and with obesity, macrosomia, and some congenital malformations involving the spine, cardiovascular system, and brain [13].

In our study, the prevalence of 16p11.2 microdeletion was approximately 1.63%, which is relatively high compared to the population prevalence previously reported in the literature. This reason is considered related to referral bias. In addition, we found that most of the prenatal microdeletions were 500–600 kb deletions in the 29.5–30.1 Mb region of chromosome 16, which is a typical 16p11.2 microdeletion (OMIM: 611913), accounting for about 85.19% (69/81) of cases. This region covers KCTD13, TBX6, HIRIP3, SEZ6L2, and other genes, consistent with previous studies [6]. In our study, among 81 fetuses diagnosed with 16p11.2 deletion, 48.15% (39/81) of the fetuses were ultrasonically identified as having hemivertebra anomalies, considering that fetal vertebral anomalies may serve as the primary intrauterine phenotype of 16p11.2 microdeletion syndrome; therefore, when a fetal ultrasound reveals hemivertebra anomalies or other skeletal malformations, CMA will be initiated immediately to exclude CNVs. This view is consistent with previous studies [8]. In our overall research cohort, CMA detected 218 cases of hemivertebra malformations, with a positive diagnostic rate of 4.39% (218/4968). In total, 39 cases of hemivertebra malformations were found in 81 fetuses diagnosed with 16p11.2 deletion, including 36 cases of pathogenic variation and three cases of likely pathogenic variation. Our study fully confirmed the important value of CMA in detecting vertebral abnormalities. In addition, the 16p11.2 microdeletion phenotype has reduced penetrance and rather variable expressivity. Possible presentations range from asymptomatic or nearly asymptomatic individuals to isolated behavioral anomalies, to autism spectrum disorder, intellectual disability, or other neurodevelopmental disorders, either isolated or associated with cases with multisystemic involvement. 16p11.2 microdeletion syndrome is the end of a wider spectrum. The multisystem malformations in 16p11.2 microdeletion syndrome include autism, intellectual disability, skeletal system malformations, and cardiovascular system malformations. Although the clinical phenotypic features associated with postnatal 16p11.2 microdeletions are well-defined, these features have not been systematically described in prenatal cases due to the limitations of prenatal identification. Therefore, targeted testing for this microdeletion prenatally is currently not feasible and due to incomplete penetrance and variable expressivity, predicting the potential phenotype of this deletion in genetic counseling is challenging. Our study is the first data on prenatal 16p11.2 microdeletion syndrome to report that 48.15% (39/81) of fetuses had skeletal malformations, 30.86% (25/81) had cardiovascular malformations, 11.11% (9/81) exhibited central nervous system malformations, 6.17% (5/81) had structural abnormalities of the digestive system, and 3.70% (3/81) had isolated ultrasound markers. These findings may provide a phenotypic basis for prenatal diagnosis of 16p11.2 microdeletion syndrome. However, our series is not representative of all fetuses with 16p11.2 microdeletions because CMA was originally offered for the presence of an ultrasound anomaly in our study.

In the present study, 14.81% (12/81) of the 81 fetuses with 16p11.2 microdeletions detected by CMA had not only skeletal or cardiovascular malformations but also had a combined NT thickening. In our center, the NT thickening standard is strictly in accordance with the standard formulated by the Fetal Medicine Foundation (FMF). NT thickening is defined as NT thickness greater than 3 mm. Among 12 fetuses with NT thickening, three cases had NT thickness exceeding 6.5 mm (case 4: 6.7 mm, case 8: 7.4 mm, case 11: 6.9 mm), and the rest of the nine cases had NT thickness between 3.0 mm and 6.4 mm. NT measurement is commonly performed between 11 and 13 weeks of pregnancy as a screening for chromosomal anomalies. Combined with the data of our current study, we suggest that when an ultrasound shows fetal NT thickening, the occurrence of chromosomal aneuploidy should not be the only factor to be considered because NT involves a variety of fetal structural abnormalities, 16p11.2 microdeletion syndrome being one of them. Therefore, when fetal NT thickening is found, we should be alert to the occurrence of chromosomal abnormalities. Fortunately, CMA can improve the detection rate of chromosome abnormalities in NT-thickened fetuses. We, therefore, propose for the first time that NT thickening may be a prenatal ultrasound indication for pCNVs in the 16p11.2 microdeletion syndrome, expanding the prenatal phenotype of this microdeletion syndrome. In addition, Lin S. et al. also demonstrated the clinical utility of CMA in fetuses with malformation or US markers [8].

In this study, 48.15% (39/81) of 81 fetuses with 16p11.2 microdeletion syndrome exhibited hemivertebra malformations, among the deletions, 31 encompassed the TBX6 gene. Al-Kateb et al. proposed that the *TBX6* gene encodes a transcription factor that is a key gene causing vertebral malformations in patients with 16p11.2 microdeletion syndrome [5,14,15]. The TBX6 (OMIM: 602427) gene, which is located on the 16p11.2 chromosome, was shown through animal studies to be crucial in congenital spinal abnormalities. The primary mechanism involves the haploinsufficiency mechanism, which leads to reduced regulation of downstream genes due to mislocalization of the T-Box transcription factor 6, thereby dysregulating the Notch signaling pathway, which is essential for somite development. Findings of Ren et al. in humans and mice consistently support that an increased TBX6 dosage contributes to the risk of developing cervical congenital vertebral malformations [16]. Pour et al. [17] found that TBX6 knockout mice showed the same phenotype of vertebral rib dysplasia and congenital scoliosis as humans. Fei et al. [15] identified the haplotype T-C-A of single nucleotide polymorphisms (SNPs) in the TBX6 gene (3 common: rs2289292, rs3809624, rs3809627) can cause hemivertebrae and scoliosis. The frequency of the T-C-A haplotype reaches 44% among Han Chinese individuals in the 1000 Genomes Project. Therefore, the allelic haplotype T-C-A of TBX6 should not be overlooked in hemivertebra and congenital scoliosis [18].

In the present study, malformations of the cardiovascular system were also part of the phenotype of this microdeletion syndrome, being identified in about 30.86% of cases (25/81), VSD being the most represented (72%, 18/25). In addition, cardiovascular malformations such as aortic constriction, aortic valve malformation, and transposition of the aorta were also found. In the cardiovascular malformations associated with 16p11.2 microdeletion syndrome, our research found that 32% (8/25) of microdeletion regions of cardiovascular malformations contain TBX1, GJA5, HIRIP3, and other genes. TBX1 is a DNA-binding protein that regulates the transcription of several genes and is involved in cardiac development and limb patterning. GJA5 is a gap junction protein that plays a key role in developing cardiac structures. The HIRIP3 gene product and HIRA bind to form the HIRA-HIRIP3 complex, which plays an important function in chromatin and histone metabolism. HIRIP3 gene haploinsufficiency was suggested as a possible association with the development of cardiac arterial valve malformations [19]. Our study is the first to report that TBX1 and CJA5 are associated with 16p11.2 microdeletion syndrome, expanding the disease spectrum of 16p11.2 microdeletion syndrome. In addition, our study suggests that VSD is the main feature of cardiac structural abnormalities caused by 16p11.2 microdeletion syndrome. 16p11.2 microdeletion syndrome can present with prenatal ultrasound abnormalities in all systems, but vertebral anomalies (48.15%) and structural cardiac anomalies (30.86%) are the most common. The presence of fetal hemivertebra anomalies, scoliosis, and structural cardiac anomalies on prenatal ultrasound in the Chinese Han population should be considered as a possible 16p11.2 microdeletion syndrome.

Neurodevelopmental abnormalities are also part of the clinical phenotype of 16p11.2 microdeletion syndrome, which mainly includes autism, intellectual disability, epilepsy, and growth retardation. This undoubtedly places a huge burden on both the family and society, so the prenatal diagnosis of the syndrome is crucial and guides the decision-making and management of the pregnancy for both the pregnant couple. In total, 6.17% (5/81) of the fetuses in this study exhibited central nervous system malformations. One case of agenesis of the corpus callosum, one case of macrocephaly, one case of hydrocephalus, and one case of an arachnoid cyst; another notable case is microcephaly. Microcephaly has not been reported in the 16p11.2 microdeletion phenotype but only in the 16p11.2 microduplication phenotype. Our study is the first time to report microcephaly in the 16p11.2 microdeletion phenotype [10]. Central nervous system malformations have different forms. The 16p11.2 microdeletion region contained the PRRT2, SEZ6L2, and KCTD13 genes in these fetuses and the mechanism by which deletion of these genes leads to the patient’s phenotype is currently unknown. However, it was suggested that the haploinsufficient dosage of the SEZ6L2 gene may be an important factor in language delay, cognitive impairment, and autism in 16p11.2 microdeletion syndrome [20]. The KCTD13 gene is a key driver of neuronal proliferation in zebrafish and mice and a major driver of the 16p11.2 microdeletion syndrome macrocephaly, and MAPK3 and MVP genes in the deletion region may act as modifier genes to enhance the expression of KCTD13 gene [21]. Prenatal ultrasound showed that the case of microcephaly gave birth to a boy at 35 + 2 weeks. At the age of 1 year, he began to show growth retardation and intellectual disability. However, there are no reports about microcephaly in 16p11.2 microdeletion syndrome; only macrocephaly was reported, and our study is the first report. We considered whether microcephaly could be considered as one of the prenatal phenotypes of 16p11.2 but this idea is a bit bold, and a larger sample size and functional research are needed to supplement this view. We propose that microcephaly can be considered for inclusion in the prenatal phenotype of 16p11.2 microdeletion syndrome to expand its prenatal phenotypic spectrum. Therefore, when a prenatal ultrasound reveals CNS abnormalities, CMA should be provided to exclude relevant pathogenic CNVs such as 16p11.2 microdeletion. Lin S. et al. [8] put forward a similar view. In addition, by following up on 23 live births, in addition to 12 cases showing growth retardation and varying degrees of intellectual disability, our study confirms the previous study that 16p11.2 microdeletion syndrome can lead to obesity. Eight cases exhibited obesity, and this region contains the SH2B1 gene, the deletion of which can lead to obesity in humans [22]. The gene is also associated with congenital renal anomalies, abnormal bladder function, and ureteral abnormalities. However, only one patient in this study was found to have polycystic kidneys in addition to exhibiting obesity. Although there is only one case, it should not be ignored. When other renal anomalies, such as polycystic kidneys, are detected on prenatal ultrasound, special attention should also be paid to the possibility that 16p11.2 microdeletion syndrome may occur, and pCNVs should be excluded by CMA.

Prenatal ultrasound structural screening is the most common, noninvasive, and repeatable method for fetal detection of congenital malformations. However, it is undeniable that ultrasound also has certain limitations. Ultrasound is influenced by various factors including maternal weight, gestational week, as well as fetal position, amniotic fluid, and fetal skeletal acoustic shadow, many organs or parts may not be displayed or may not be displayed well. It is also impossible to show all the structures of the fetus on ultrasound, and some fetal anomalies are very difficult or even impossible to diagnose on prenatal ultrasound. Prenatal ultrasound and CMA are complementary, reducing the birth rate of congenital malformations.

Although this study is the largest data study of 16p11.2 microdeletion syndrome to date, it is also the first to propose NT thickening and polycystic kidney as prenatal ultrasound phenotypes of 16p11.2 microdeletion syndrome, expanding the disease spectrum of 16p11.2 microdeletion syndrome. However, there are still some limitations to this study. Firstly, we did not perform functional research experiments and could not provide a more in-depth description of the genes involved. Secondly, our assay may have missed balanced chromosomal rearrangements. Thirdly, only 23 fetuses were born in the 81 cases of pCNVs detected by CMA, and the follow-up on post-natal conditions may be biased. Fourthly, our cohort is NOT representative of all fetuses with 16p11.2 deletion but only of those initially presenting with ultrasound anomalies and later diagnosed with 16p11.2 microdeletion by CMA. This means the study is very useful in defining a possible prenatal presentation for this condition but should be taken cautiously when providing a prognosis in future cases. Lastly, a further limitation is that our research lacks sequencing data. In some cases, the association of this CNV with the phenotype (e.g., some renal anomalies and some central nervous system anomalies) is not established. Whole Exome Sequencing should be encouraged to confirm or botch the supposed association.

## 5. Conclusions

In summary, in our cohort, the most common ultrasound findings in 16p11.2 microdeletion fetuses were skeletal malformations (especially hemivertebra malformations), followed by cardiovascular malformations (VSDs were most frequent) and isolated ultrasound markers. Our findings enriched the prenatal ultrasound phenotype of 16p11.2 microdeletion, which will be useful for genetic counseling in clinical settings. We hope that the information provided in this study will contribute to more effective genetic counseling and professional clinical management of pregnant couples with prenatal detection of 16p11.2 microdeletion syndrome and reduce the occurrence of adverse pregnancy outcomes. In addition, pre-implantation genetic testing should be a first-tier method for such carriers if they plan to conceive again.

## Figures and Tables

**Table 1 genes-13-02315-t001:** Clinical details of 81 fetuses with 16p11.2 microdeletion in the study.

Maternal age (median)	30.1 (range 19–47) years
Gestational weeks (median) ^A^	25.8 (range 12–37) weeks
Sex of fetuses (M/F)	29/52
Sample types	
Amniotic fluid	57
Cord blood	24
Malformation classification	
Isolated	3
Non-isolated	78
Follow-up	23

^A^ Means the gestational age at del 16p11.2 diagnosis.

**Table 2 genes-13-02315-t002:** Classification of abnormal ultrasound findings in 81 fetuses with 16p11.2 microdeletion.

Ultrasound Finding	*n* (%, n/N)
Multiple malformations	6 (7.41,6/81)
Isolated malformations	3 (3.70, 3/81)
Skeletal system	39 (48.15, 39/81)
Vertebral malformations	31 (38.27, 31/81)
Talipes equinovarus	3 (3.70, 3/81)
Craniosynostosis	2 (2.47, 2/81)
Limb anomaly	1 (1.23, 1/81)
Congenital scoliosis	2 (2.47, 2/81)
Cardiovascular system	25 (30.86, 25/81)
Ventricular/atrial septal defect	13 (16.05, 13/81)
Supravalvular aortic stenosis/coarctation	3 (3.70, 3/81)
Tricuspid insufficiency	2 (2.47, 2/81)
Double outlet right ventricle	1 (1.23, 1/81)
Endocardial cushion defect	1 (1.23, 1/81)
Hypertrophic cardiomyopathy	1 (1.23, 1/81)
Pulmonic stenosis	2 (2.47, 2/81)
Central nervous system	9 (11.11, 9/81)
Dandy–Walker malformation	1 (1.23, 1/81)
Agenesis of the corpus callosum	3 (3.70, 3/81)
Microcephaly	2 (2.47, 2/81)
Vein of Galen aneurysm	1 (1.23, 1/81)
Spinal Bifida	1 (1.23, 1/81)
Cephalocele	1 (1.23, 1/81)
Digestive system	5 (6.17, 5/81)
Choledochal cyst	2 (2.47, 2/81)
Duodenal obstruction	3 (3.70, 3/81)
Others	12 (14.81, 12/81)
Lymphatic cyst	5 (6.17, 5/81)
Hydronephrosis	3 (3.70, 3/81)
Arteriovenous malformations	1 (1.23,1/81)
Fetal growth restriction	2 (2.47, 2/81)
Tumor	1 (1.23, 1/81)

**Table 3 genes-13-02315-t003:** The perinatal outcomes in fetuses with 16p11.2 microdeletion.

	Isolated ^a^ vs. Non-Isolated ^b^				Genetic Testing Results			
Perinatal Outcome	Total (*n* = 4968)	Isolated (*n* = 3936)	Non-Isolated (*n* = 3936)	*p*-Value	CNVs (*n* = 81)	VUS (*n* = 18)	Negative (*n* = 4869)	*p*-Value
TOP	438(8.82%)	101(2.57%)	337(32.66%)	0.000	56(69.14%)	5(27.78%)	377(7.74%)	0.000
Live birth	4492(90.42%)	3826(97.21%)	666(64.53%)	0.000	23(28.39%)	12(66.67%)	4457(91.54%)	0.000
Neonatal death	38(0.76%)	9(0.23%)	29(2.81%)	0.000	2(2.47)	1(5.56%)	35(0.72%)	0.490

CNVs: copy number variants; VUS: variants of unknown significance; TOP: termination of pregnancy. ^a^ Fetus with only one ultrasound abnormality marker. ^b^ Fetus with at least two or more ultrasound abnormalities markers.

## Data Availability

The original contributions presented in the study are included in the article, further inquiries can be directed to the corresponding author.

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
