# Peer review of "Prenatal Diagnosis of Chromosome 16p11.2 Microdeletion"

_genes, 2022, doi:10.3390/genes13122315_

Round 1
Reviewer 1 Report
Included in the document upload

Author Response
Dear Editor and Reviewers,
Thank you very much for giving us an opportunity to revise our manuscript entitled, " Prenatal diagnosis of chromosome 16p11.2 microdeletion" Genes-2047600. We appreciate the time and effort that you and the reviewers dedicated to providing feedback on our manuscript and are grateful for the insightful comments on and valuable improvements to our paper. We have carefully studied the reviewer's comments carefully and tried our best to revise according to the comments. The language editing is carried out by an English expert. Revised portions are marked in red in the revised paper.
We have incorporated most of the suggestions made by the reviewers. Those changes are highlighted within the manuscript. Please see below for a point-by-point response to the reviewers’comments and concerns. All page numbers refer to the revised manuscript file with tracked changes.
Thank you very much for your attention and consideration.
We would like also to thank you for allowing us to resubmit a revised copy of the manuscript.
We hope that the revised manuscript is accepted for publication in Genes.
Institute: The first clinical medical college, Southern Medical University, Guangzhou, China
Tel: +86 (0)20 38076346
Fax: +86 (0)20 38076337
E-mail: You Wang, wy13781539630@163.com; Can Liao, canliao6008@163.com
Sincerely yours
Dr. You Wang

Author Response
Dear Editor and Reviewers,
Thank you very much for giving us an opportunity to revise our manuscript entitled, " Prenatal diagnosis of chromosome 16p11.2 microdeletion" Genes-2047600. We appreciate the time and effort that you and the reviewers dedicated to providing feedback on our manuscript and are grateful for the insightful comments on and valuable improvements to our paper. We have carefully studied the reviewer's comments carefully and tried our best to revise according to the comments. The language editing is carried out by an English expert. Revised portions are marked in blue in the revised paper.
We have incorporated most of the suggestions made by the reviewers. Those changes are highlighted within the manuscript. Please see below, for a point-by-point response to the reviewers’comments and concerns. All page numbers refer to the revised manuscript file with tracked changes.
Thank you very much for your attention and consideration.
We would like also to thank you for allowing us to resubmit a revised copy of the manuscript.
We hope that the revised manuscript is accepted for publication in Genes.
Institute: The first clinical medical college, Southern Medical University, Guangzhou, China
Tel: +86 (0)20 38076346
Fax: +86 (0)20 38076337
E-mail: You Wang, wy13781539630@163.com; Can Liao, canliao6008@163.com
Sincerely yours
Dr. You Wang

Reviewer 3 Report
Rows 18-20 – sentence ”Chromosome microarray analysis (CMA) was performed on them, and the prenatal ultra- 18 sound characteristics and genetic testing results of 81 fetuses diagnosed with 16p11.2 microdeletion 19 were recorded and evaluated pregnancy outcomes.” – requests reformulation.
Rows 21-24 – sentence ”Which, 48.15% (39/81) of fetuses had skeletal mal- 21 formations, 30.86% (25/81) had cardiovascular malformations, 11.11% (9/81) showed central nervous 22 system malformations (CNS), 6.17% (5/81) had structural abnormalities of the digestive system, and 23 3.70% (3/81) had isolated ultrasound markers.” – requests reformulation.
Rows 71-72 – sentence ”Discard CMA if QF-PCR results show aneuploidy. Karyotype analysis using con- 71 ventional G-banding technology (550-band resolution).” – requests reformulation.
Row 85 – sentence ”Complete statistical analysis using the IBM statistical program SPSS 26.0.” – requests reformulation.
Rows 91-97 – sentences ”A total of 75 pathogenic variants, 6 likely pathogenic variants, and 18 VOUS were detected for copy number deletion in the 16p11.2 region. Of these 81 pregnancies, 56 were menstrual, and 25 were primigravida. At the initial ultrasound evaluation, Gestational age ranged from 11 to 31 weeks and between 17 and 37 weeks for invasive prenatal genetic testing. Amniotic fluid blood samples were obtained in 65 cases, and cord blood samples were obtained in 16 cases for CMA and G manifest band karyotyping.” – request reformulation.
Rows 103-105 – sentence ”Of these, 14.81% (12/81) had a combination of nuchal translucency (NT) thickening and 3.70% (3/81) had isolated ultrasound markers (Table 1).” – request reformulation.
Row 105 – the elements presented in the sentence was mentioned Table 1 have nothing to do with those presented in the Table 1
Rows 110-112 – sentence ”The fetus presents with malformations of the cardiovascular system, including ventricular septal defect (VSD), atrial septal defect (ASD), aortic stenosis and prolapse, endocardial cushion defect, and pulmonary stenosis.” – requests reformulation.
Rows 110-112 – sentence ”Notably, in 30.86% of fetuses with structural malformations of the cardiovascular system, skeletal malformations were observed successively in 4 fetuses at 32-37 weeks, all exhibiting hemivertebra malformations.” – requests reformulation.
Rows 122-124 – sentences ”8 chromosomal abnormalities were detected by QF-PCR, including 1 case of XXY with sex chromosome abnormalities. By CMA, confirmed 2 cases of trisomy 21, 4 cases of 123 trisomy 16, and 1 case of trisomy 13.” – these sentences are inutile, request the elimination.
Rows 139-141 – sentences”4 cases of unilateral clubfoot and one case of bilateral clubfoot were not identified by prenatal ultrasound but were identified during screening in 23 newborns.” – request reformulation.
Table 2 contains the elements that are not mentioned in the text. The suggestion is to mention in the text the data concerning microduplication 16p11.2 and also to detail what means ”others”.
Table 3 and Table 4 are not mentioned in the text.
In table 4 is not clear what means ”isolated vs. non-isolated”
Sentence (rows 165-167) ”16p11.2 microdeletion refers to the ~600kb heterozygous deletion between BP4 and BP5 between 29.5 and 30.1Mb on chromosome 16 in the reference genome (GRCh37/hg19)[11].” it is not clear; the deletion covers 600kb or 30Mb?
The sentence (rows 165-167): ”In addition, this deletion has been associated with other mental or neurological disorders (such as speech delay, mental retardation, developmental coordination problems, and epilepsy) and with obesity, macrosomia, and some congenital malformations involving the spine, cardiovascular system, and brain.” requests a bibliographic indication.
The sentence (row 179) ”this reason is considered related to referral bias.” requests caps lock at the start.
The use of term ”incomplete epistasis” (row 193) is incorrect; probably the authors vised the ”incomplete penetrance”.
The sentence ”Our study is the first and largest data to report that 48.15% (39/81) of fetuses had skeletal malformations, 30.86% (25/81) had cardiovascular malformations, 11.11% (9/81) exhibited central nervous system malformations, 6.17% (5/81) had structural abnormalities of the digestive system, and 3.70% (3/81) had isolated ultrasound markers” (rows 195-199) is incomplete; it is requested to refer to the microdeletion 16p.
The sentence ”In this study, 48.15% (39/81) of 81 fetuses with 16p11.2 microdeletion syndrome exhibited hemivertebra malformations, 31 of which contained the TBX6 gene.” (rows 210-211) requests reformulation; who contains TBX6 gene?
In sentences ” SPARROW et al. [14]demonstrated that mutations in the TXB6 gene affect the transcriptional activity of the TBX6 protein, resulting in vertebral rib dysplasia, and found that 46% of mouse embryos carrying a nonfunctional allele of TBX6 had mild cervical defects and 30% had sacral defects. POURQUIE et al. [15]found that TXB6 knockout mice showed the same phenotype of vertebral rib dysplasia and congenital scoliosis as humans. FEI et al. [16]identified two single nucleotide polymorphisms in the TBX6 gene: rs2289292 and rs3809624, which play an important role in the pathogenesis of congenital scoliosis in the Chinese Han population.” (rows 213-220) I think that the use of caps lock is inutile; in addition after the square brackets is requested a space.
The sentence ”We are concerned that 32% (8/25) of microdeletion regions of cardiovascular malformations contain TBX5, GJA5, HIRIP3, and other genes.” (rows 227-228) is not clear.
In sentence ”The HIRIP3 gene product and HIRA bind to form the HIRA- HIRIP3 complex, which plays an important function in chromatin and histone metabolism. HIRIP3 gene haploinsufficiency has been suggested as a possible association with the development of cardiac arterial valve malformations[17].” (rows 231-234) before the square brackets is requested a space.
In sentence ”The KCTD13 gene is a key driver of neuronal proliferation in zebrafish and mice and a major driver of the 16p11.2 microdeletion syndrome macrocephaly, and MAPK3 and MVP genes in the deletion region may act as modifier genes to enhance the expression of KCTD13 gene[18].” (rows 253-256) before the square brackets is requested a space.
In sentence ” Eight cases exhibited overweight, and this region contains the SH2B1gene, the deletion of which can lead to obesity in humans[19].” (rows 267-268) before the square brackets is requested a space.
Sentence ”In summary, our study highlights the application of CMA in 16p11.2 microdeletion syndrome.” (rows 298-299) seems to be incomplete; I suggest to add ”diagnosis” after syndrome.
All genes abbreviations must be presented in italic.
Author Response
Dear Editor and Reviewers,
Thank you very much for giving us an opportunity to revise our manuscript entitled, " Prenatal diagnosis of chromosome 16p11.2 microdeletion" Genes-2047600. We appreciate the time and effort that you and the reviewers dedicated to providing feedback on our manuscript and are grateful for the insightful comments on and valuable improvements to our paper. We have carefully studied the reviewer's comments carefully and tried our best to revise according to the comments. The language editing is carried out by an English expert. Revised portions are marked in purple in the revised paper.
We have incorporated most of the suggestions made by the reviewers. Those changes are highlighted within the manuscript. Please see below, for a point-by-point response to the reviewers’comments and concerns. All page numbers refer to the revised manuscript file with tracked changes.
Thank you very much for your attention and consideration.
We would like also to thank you for allowing us to resubmit a revised copy of the manuscript.
We hope that the revised manuscript is accepted for publication in Genes.
Institute: The first clinical medical college, Southern Medical University, Guangzhou, China
Tel: +86 (0)20 38076346
Fax: +86 (0)20 38076337
E-mail: You Wang, wy13781539630@163.com; Can Liao, canliao6008@163.com
Sincerely yours
Dr. You Wang

Round 2
Reviewer 1 Report
no comments
Reviewer 3 Report
This version is OK and in my opinion other changes are not necessary.